# DFT Studies on the Antioxidant Activity of Naringenin and Its Derivatives: Effects of the Substituents at C3

**DOI:** 10.3390/ijms20061450

**Published:** 2019-03-22

**Authors:** Yan-Zhen Zheng, Geng Deng, Rui Guo, Da-Fu Chen, Zhong-Min Fu

**Affiliations:** 1College of Bee Science, Fujian Agriculture and Forestry University, Fuzhou 350002, China; yanzhenzheng@fafu.edu.cn (Y.-Z.Z.); rui_0508@163.com (R.G.); fzmfafu126@126.com (Z.-M.F.); 2Key Laboratory of Bioorganic Phosphorous Chemistry and Chemical Biology (Ministry of Education), Department of Chemistry, Tsinghua University, Beijing 100084, China; dengg@mail.tsinghua.edu.cn

**Keywords:** naringenin, substituent effect, structure-antioxidant activity relationship, Hammett sigma constants, density functional theory

## Abstract

The radical scavenging activity of a flavonoid is largely influenced by its structure. The effects of the substituents at C3 position on the antioxidant activity of naringenin were carried out using the density functional theory (DFT) method. The reaction enthalpies related with the three well-established mechanisms were analyzed. Excellent correlations were found between the reaction enthalpies and Hammett sigma constants. Equations obtained from the linear regression can be helpful in the selection of suitable candidates for the synthesis of novel naringenin derivatives with enhanced antioxidant properties. In the gas and benzene phases, the antioxidant activity of naringenin was enhanced by the electron-donating substituents via weakening the bond dissociation enthalpy (BDE). In the water phase, it was strengthened by electron-withdrawing groups—via lowering the proton affinity (PA). The electronic effect of the substituent on the BDE of naringenin is mainly governed by the resonance effect, while that on the ionization potential (IP) and PA of naringenin is mainly controlled by the field/inductive effect.

## 1. Introduction

Excess formation of reactive free radicals by various enzymatic and non-enzymatic processes in the body has been found associated with the oxidation of biomolecules such as nucleic acids, proteins and lipids [1,2]. The oxidations of biomolecules are closely related with the initiation and progression of various diseases such as atherosclerosis, cardiovascular disease, neurodegenerative disease, cancer, and aging [1,3].

Antioxidants are chemical compounds that can trap free radical intermediates formed during the oxidative reactions and inhibit the oxidative reactions. The natural antioxidants have recently received notable attention because they can avoid or at least significantly reduce the oxidation of biomolecules by free radicals [4,5]. Among the multiple natural bioactive compounds, flavonoids as free radical chain-breaking antioxidants are well known. Flavonoids are a class of naturally occurring polyphenolic compounds that exist in a wide variety of foods and thus are important constituents of the human diet [6]. They are widely found in natural foods such as fruits, vegetables, cereals, teas, wines, honey, and bee pollen [6].

The flavonoid basic structure consists of two benzene rings (A and B) connected by an oxygen-containing pyrene ring (C) (Figure 1). To date, more than 6000 flavonoids have been identified. Flavonoids possess a broad range of biological activities including antibacterial, antiviral, anti-inflammatory and anti-ischemic activities [7,8,9,10], which are closely related with their free radical scavenging activities. It is reported that flavonoids play their anti-inflammatory action mainly via the ability to modulate production of free radical by phagocytic leukocytes [11]. Naringenin (4′,5,7-trihydroxyflavanone, Figure 1) is a dietary flavonoid abundant in foods such as citrus fruits, honey, and bee pollen. Naringenin has attracted increasing attention due to its positive health benefits in the human body. It has been reported to have several biological effects, such as monoamine oxidase inhibitory [12] and neuroprotective activities [13]. Monoamine oxidase inhibitory and neuroprotective activities of naringenin may provide protection against oxidative neurodegeneration and alleviate central nervous system disorders such as depression [14].

The antioxidant activity of phenolic compounds depends on their capacity to resist the detrimental effect of free radicals. There is a close relationship between antioxidant activity and structural properties. Among multiple structural features, the substituent effect is one of the most important factors that influence the antioxidant properties of phenolic compounds [15,16,17,18]. The studies on the influence of the substituent effect can be used to synthesize and select novel compounds with better antioxidant activity. Until now, there have been multiple studies that focus on the number and position of hydroxyl groups in relation to the biological activity of the flavonoids [19,20,21]. On the other hand, the substituent effect on the antioxidant activity of flavonoids is seldom known.

There are several mechanisms for the free radical scavenging action of phenolic compounds and the net result of all mechanisms is the same: transferring a hydrogen atom to the free radicals. The following mechanisms are widely accepted for phenolic compounds exerting their antioxidant activity [22,23,24]: (1) the hydrogen atom transfer (HAT) in which the radical abstracts the hydrogen by homolytic cleavage of the hydroxyl group (Equation (1)); (2) the single-electron transfer followed by proton transfer (SET-PT) takes place through two consecutive steps: an electron donates from the phenolic compounds (the first step, Equation (2)) following by a proton transferring from the formed cation radical (the second step, Equation (3)); (3) the sequential proton loss electron transfer (SPLET), which takes place in two steps: a proton transfers from the antioxidant (the first step, Equation (4)) following by an electron donating from the anion created in the first step (the second step, Equation (5)).

R• + ArOH → RH + ArO•(1)

R• + ArOH → R^−^ + ArOH^+^•(2)

R^−^ + ArOH^+^• → RH + ArO•(3)

ArOH → ArO^−^ + H^+^(4)

ArO^−^ + R• + H^+^ → ArO• + RH(5)

Understanding the role of substituent is of great importance for the preparation of novel phenolic compounds with enhanced antioxidant property. The computational method is a powerful tool that can provide valuable information about the antioxidant character of the phenolic compound. The work that is difficult to carry out by the experiment method can be performed by the computational method. Computational methods, especially the density functional theory (DFT), have been successfully used to evaluate the chemical properties, which are closely related with the antioxidant activity [22,23,24,25,26,27,28,29,30,31,32,33,34,35,36]. In this paper, the effects of various substituents on the antioxidant activity of naringenin were investigated by the DFT method. Reaction enthalpies related to HAT, SET-PT, and SPLET mechanisms were calculated. The derivatives of flavonoids with substituents at C3 positions are widely distributed in natural foods such as honey and bee pollens. In this work, various substituents (NH_2_, OMe, Me, OH, F, Cl, CHO, CF_3_, CN and NO_2_) covering the electron-withdrawing groups and electron-donating groups were placed at the C3 position of naringenin. The Hammett sigma constant has been one of the most widely used means to study and interpret the organic reactions and their mechanisms [18]. To find a relationship between substituents and the antioxidant activity, correlations of calculated reaction enthalpies with Hammett sigma constants of the substituents were investigated.

## 2. Results and Discussion

The atom numberings for the carbon atoms in the basic structure of a flavonoid are shown in Figure 1. For simplicity, the numberings for oxygen and hydrogen atoms are used to correspond to the number of attached carbon atoms. For example, atoms of OH group at C5′ are O5′ and H5′, at C3 are O3 and H3. The naringenin monomer is used as the same atom numbering as the flavonoid basic structure.

Reaction enthalpy characterizing the first step of the reaction mechanism is of great importance in evaluating the antioxidant action. In the three mechanisms, HAT is a one-step mechanism, while SET-PT and SPLET mechanisms take place via two steps. Hence, the following analyses mainly focus on the bond dissociation enthalpy (BDE), ionization potential (IP) and ionization potential (PA) characterizing the first steps of the three mechanisms.

### 2.1. HAT Mechanism

#### 2.1.1. Calculated BDEs of Naringenin and its C3-Substituted Derivatives

In a HAT mechanism, a hydrogen atom abstracts from the phenolic hydroxyl group to the free radical via homolytic cleavage. In this mechanism, BDE is the parameter evaluating the antioxidant action of a compound. The lower the BDE, the stronger the antioxidant activity and the more important the role of the corresponding O−H group in the antioxidant action. The calculated O−H BDEs of naringenin and its C3-substituted derivatives are given in Table 1. The antioxidant activities of naringenin and its C3-OH substituted derivatives kaempferol (4′,3,5,7-tetrahydroxyflavanone) have been reported in previous work [37]. It is found that in the aqueous phase, the antioxidant activity of kaempferol is stronger than naringenin [37]. In this work, the BDEs of naringenin in different hydroxyl group are stronger than kaempferol in the water phase. It is consistent with the experimental results, thus, confirming the accuracy of our work.

For naringenin and its derivatives, they have a common feature: the lowest BDEs are at 4′−OH irrespective of the studied media. Therefore, the antioxidant capacities of naringenin and its derivatives are preferred to present by 4′−OH in the HAT mechanism. This observation may be attributed to that of the flavonoiud−O4′, where the radical is more stable in the studied phases.

ΔBDE representing the difference between BDE of the substituted naringenin and naringenin itself, reflects the effect of the substituents on O–H BDE. The calculated ΔBDEs are summarized in Appendix A. In the gas phase, the lowest effect of the substituent on the O-H BDE is found for Me group at C3 position, where BDEs vary only by 0.5, −0.8 and −0.2 kJ/mol for 4′-OH, 5-OH and 7-OH, respectively. The biggest effect of the substituent on the O-H BDE is found for NO_2_ group at C3 position, where BDEs vary by 7.2, 4.4, and 6.2 kJ/mol for 4′-OH, 5-OH, and 7-OH respectively. The similar trends are also observable in the benzene and water phases. Thus, the Me group at C3 position influences the antioxidant activity of naringenin weaker compared with other groups, while the NO_2_ group has the strongest impact on the antioxidant action.

#### 2.1.2. Dependence of BDEs on Hammett Sigma Constants

The Hammett sigma constant has been one of the most widely used means to study and interpret the organic reactions and their mechanisms [18]. In this work, the Hammett sigma constants: *σ*_m_ and *σ*_p_ of the substituents were used. Appendix A lists the *σ*_m_ and *σ*_p_ of the studied substituents collected by Hansch et al. [18]. They were obtained from the ionization of organic acids in solutions and have been successfully used to predict the equilibrium and rate constants of a family variety of reactions. To see the correlation of the Hammett sigma constants with the BDE of the substituted naringenin, the Pearson correlation was carried out. The *P*(BDEs, *σ*) are drawn as a histogram in Figure 2. In Figure 2, all of the *P*(BDEs, *σ*) are positive and larger than 0.8. Hence, the BDEs are positively and highly relevant with the Hammett sigma constants. The positive coefficients also character that electron-withdrawing groups at C3 position raise the BDE of naringenin, while electron-donating groups play the contrary role. The obtained results can be interpreted that electron-donating groups at the C3 position can stabilize the radical, while destabilizing the parent molecule. Hence, they decrease the O−H BDE. However, electron-withdrawing groups at the C3 position have the opposite effect, and their presence would lead to an increase in the O−H BDE. On the other hand, the electron-donating groups placed at the C3 position would strengthen the antioxidant activity of naringenin, while the electron-withdrawing groups have an opposite effect in a HAT mechanism.

In Figure 2, the *P*(BDEs, *σ*_p_) are larger than *P*(BDEs, *σ*_m_) and exceed 0.9. Therefore, the correlation between the BDE of naringenin and σ_p_ is better than that between the BDE and *σ*_m_ and the BDE of naringenin can be predicted better by *σ*_p_. The equations obtained from the linear regressions between *σ*_p_ and BDE(4′−OH) are placed in the Figure 3A. They can be used to predict the BDE(4′−OH) by the Hammett sigma constants. As the 4′−OH is the strongest antioxidant hydroxyl group, the antioxidant activity of the substituted naringenin can also be predicted by the obtained equations.

#### 2.1.3. The Electronic Effects of the Substituents on the BDEs

In this work, the electronic effects of the substituents on the BDEs have been considered. The electronic effect of the substituent is mainly composed of two parts: field/inductive effect represented by parameter *F* and resonance effect characterized by parameter *R* [18,38,39]. The Pearson correlation coefficients between the *F*/*R* and the BDEs are drawn as a histogram in Figure 2. In Figure 2, it can be seen that the correlation between BDE of naringenin and *R* is much better than that between BDE and F. The *P*(BDEs, *R*) exceeds 0.8. Therefore, the electronic effect of substituent at the C3 position on the BDE of naringenin is mainly governed by the resonance effect.

### 2.2. SET-PT Mechanism

#### 2.2.1. Calculated IPs of Naringenin and its C3-Substituted Derivatives

Another important mechanism of the antioxidant process is SET-PT. The first step of this mechanism is transferring a single electron from a natural flavonoid. This step is significantly important for the SET-PT mechanism. It is characterized by IP. The lower the IP, the stronger the antioxidant activity. The calculated IPs for naringenin and its derivatives are given in Table 2. It is well known that the charged species are especially sensitive to the polarity change of the solvents. Thus, the IPs of naringenin and its derivatives change drastically with the increasing polarity of the environments, as can be seen in Table 2. The cause may be that the cation radicals of naringenin and its derivatives are more stable, and the delocalization and conjugation of the π-electrons are more delocalized in the polar environments.

ΔIP representing the difference between IP of the substituted naringenin and the naringenin itself reflects the effect of substituent on IP. The calculated ΔIPs are summarized in Appendix A. In the gas phase, the lowest effect of the substituent on the IP is found for Me group at C3 position, where IP varies only by 3.3 kJ/mol. The biggest effect of the substituent on the IP is found for NO_2_ group at C3 position, where IP changes by 41.4 kJ/mol. Similar trends are also observable in the benzene and water phases. Thus, the Me group at C3 position influences the antioxidant activity weaker compared with other groups, while the NO_2_ group has the strongest impact on the antioxidant action of naringenin.

#### 2.2.2. Dependence of IPs on Hammett Sigma Constants

To see the correlation of the Hammett sigma constants with the IP of substituted naringenin, the Pearson correlation were carried out. The Pearson correlation coefficients between the *σ*_m_/*σ*_p_ and the IPs are drawn as a histogram in Figure 2. In Figure 2, all of the *P*(IPs, *σ*) are positive and larger than 0.8. Hence, the IPs are positively and highly relevant with the Hammett sigma constants. The positive coefficient also characterizes that electron-donating groups at the C3 position reduce the IP of naringenin, while electron-withdrawing groups have the opposite effect. The obtained results can be interpreted that electron-donating groups at the C3 position can stabilize the cation radical and destabilize the parent molecule. Hence, they reduce the IPs. However, electron-donating groups at the C3 position have the opposite effect, and their presence would lead to an increase in the IP. Therefore, the electron-donating groups placed at the C3 position would enhance the antioxidant activity of naringenin, while the electron-withdrawing groups have an opposite effect in SET-PT mechanism.

In Figure 2, the *P*(IPs, *σ*_m_) are larger than *P*(IPs, *σ*_p_) and exceed 0.9. Therefore, the correlation between the IP in naringenin and σm is better than that between the IP and *σ*_p,_ and the IP of naringenin derivatives with the substituent at C3 position can be predicted well by *σ*_m_. The equations obtained from the linear regressions between *σ*_m_ and IP are placed in Figure 3B. They can be used to predict the IP and antioxidant activity of the substituted naringenin by the *σ*_m_. This can be useful in the selection of suitable candidates for the synthesis of novel naringenin derivatives with enhanced antioxidant properties.

#### 2.2.3. The Electronic Effects of the Substituents on the IPs

In this work, the electronic effects of the substituents on the IPs have been considered. The Pearson correlation coefficients between the *F*/*R* and the IPs are drawn as a histogram in Figure 2. In Figure 2, it can be seen that the correlation between IP of naringenin and *F* is much better than that between IP and R. The *P*(IPs, *F*) exceed 0.9. Therefore, the electronic effect of substituent at the C3 position on the IP of naringenin is mainly governed by the field/inductive effect.

### 2.3. SPLET mechanism

#### 2.3.1. Calculated PAs of Naringenin and its C3-Substituted Derivatives

In polar phases, SPLET mechanism plays an important role in scavenging free radicals of the phenolic compounds. The heterolytic O‒H bond dissociation is the first step of this mechanism. This step is significantly important for SPLET mechanism and is characterized by PA. The lower the PA, the stronger the antioxidant activity and the more important the role of the corresponding O−H bond in the antioxidant action. The calculated O−H PAs of naringenin and its derivatives are given in Table 3. The antioxidant activities of naringenin and its C3-OH substituted derivatives kaempferol (4′,3,5,7-tetrahydroxyflavanone) have been reported in previous work [37]. It is found that in the aqueous phase, the antioxidant activity of kaempferol is stronger than naringenin [37]. In this work, the PAs of naringenin in a different hydroxyl group are stronger than those kaempferol in the water phase. It is consistent with the experimental results, thus, confirming the accuracy of our work.

For naringenin and its derivatives, they have common features: the lowest PAs are at 7−OH irrespective of the studied media. Therefore, the antioxidant capacities of naringenin and its derivatives are preferred to present by 7−OH in the SPLET mechanism. This observation may be attributed to the fact that the acid strength of 7−OH is stronger and the flavonoiud−O7^−^ is more stable in the studied phases.

The charged species are especially sensitive to the polarity change of the solvents and the solvation enthalpies of the proton and anion are relatively high. Thus, from the gas phase to the solvent phases, the PAs decrease drastically. For example, the average deviation of PA(4′−OH) between the gas phase and the water phase reaches 1244 kJ/mol. This result indicates that the first step of the SPLET mechanism for naringenin and its derivatives are more favored in the polar phases.

ΔPA representing the difference between PA of the substituted naringenin and naringenin itself, reflects the effect of substituent on O–H PA. The calculated ΔPAs are summarized in Appendix A. In the gas phase, the lowest effect of the substituent on the O-H PA is found for Me group at C3 position, where PAs vary only by 0.6, 2.0 and 2.9 kJ/mol for 4′-OH, 5-OH and 7-OH, respectively. The biggest effect of the substituent on the O-H PA is found for NO2 group at C3 position, where PAs vary by −8.5, −45.7 and −38.4 kJ/mol for 4′-OH, 5-OH and 7-OH respectively. The similar trends are also observable in the benzene and water phases. Thus, the Me group at C3 position influences the antioxidant activity of naringenin weaker compared with other groups, while the NO_2_ group has the strongest impact on the antioxidant action.

#### 2.3.2. Dependence of PAs on Hammett Sigma Constants

To see the correlation of the Hammett sigma constants with the PAs of substituents naringenin, the Pearson correlation were carried out. The Pearson correlation coefficients between the *σ*_m_/*σ*_p_ and the PAs are drawn as a histogram in Figure 2. In Figure 2, all of the *P*(PAs, *σ*) are negative and the absolute value of *P*(PAs, *σ*) are larger than 0.8. Hence, the PAs are negatively and highly relevant with the Hammett sigma constants. The negative value of the coefficient also characterizes that electron-donating groups at C3 position of naringenin increase the PA, while electron-withdrawing groups have an opposite effect. The obtained results can be interpreted that electron-donating groups at the C3 position of naringenin can stabilize the neutral molecule and destabilize the anion. Hence, they increase the O−H PA. However, electron-withdrawing groups in the C3 position have the opposite effect, and their presence would lead to a decrease in the O−H PA. Therefore, the electron-donating groups placed at the C3 position would lower the antioxidant activity of naringenin while the electron-donating groups have an opposite effect in SPLET mechanism.

In Figure 2, the *P*(PAs, *σ*_m_) are larger than *P*(PAs, *σ*_p_) and exceed 0.9. Therefore, the correlation between the PA of naringenin and *σ*_m_ is better than that between the PA and *σ*_p_ and the PA of O−H can be predicted better by *σ*_m_. The equations obtained from the linear regressions between *σ*_m_ and PA(7−OH) are placed in the Figure 3C. They can be used to predict the PA(7−OH) by the Hammett sigma constants. As the 7−OH is the strongest antioxidant hydroxyl group in SPLET mechanism, the antioxidant activity of the substituted naringenin can also be predicted by the equations.

#### 2.3.3. The Electronic Effects of the Substituents on the PAs

The Pearson correlation coefficients between the *F*/*R* and the PAs are drawn as a histogram in Figure 2. In Figure 2, it can observe that the correlation between PA and F is much better than that between PA and R. The *P*(PAs, *F*) exceeds 0.8. Therefore, the electronic effect of the substituent at the C3 position on the PA of naringenin is mainly governed by the field/inductive effect.

### 2.4. The antioxidant Activity Strength Influenced by the Substituted Groups

In general, free energy (∆_r_*G* = ∆_r_*H* − *T*∆_r_*S*) is the criterion to distinguish the thermodynamically favored process. However, in the case of the studied mechanisms, the absolute values of the entropic term (−*T*∆_r_*S*) reach only few units of kJ/mol and the free energies are only shifted in comparison to the corresponding enthalpies (∆_r_*H*).

Reaction enthalpy characterizing the first step of reaction mechanism is of great importance in evaluating the antioxidant action. Thus, the BDE, IP and PA can be used to determine the thermodynamically preferred reaction pathway for the antioxidant reaction of naringenin and its C3-substituted derivatives. Analyzed the data in Table 1, Table 2 and Table 3, it can be deduced that the BDEs of 4′−OH (the strongest antioxidant hydroxyl group following the HAT mechanism) are significantly lower than the IPs and the PAs of 7−OH (the strongest antioxidant hydroxyl group following the SPLET mechanism) for naringenin and its derivatives in the gas and benzene phases. In the water phase, the PAs of 7−OH are significantly smaller than the lowest BDEs of 4′−OH and IPs. Hence, from the thermodynamic point of view, HAT is the most favored mechanism in the gas and benzene phases, while SPLET is more preferred than HAT and SET-PT in the water phase.

In the studied phases, the calculated IPs for naringenin and its derivatives are significantly higher than the lowest BDEs and the lowest PAs. Therefore, from the thermodynamic point, SET-PT is the least favored mechanism in the studied phases.

In HAT and SPLET mechanisms, the antioxidant activities of flavonoids are characterized by the lowest BDE and lowest PA, respectively. In the gas and benzene phases, the electron-donating substituents placed at the C3 position would enhance the antioxidant activity of naringenin via lowering the BDE of 4′−OH, while the electron-withdrawing groups have the opposite effect. In water phase, the electron-donating substituents placed at the C3 position would weaken the antioxidant activity of naringenin via enlarging the PA of 7−OH, while the electron-withdrawing groups have an opposite effect. Among the studied compounds, the NH_2_ substituted derivative is the strongest antioxidant in the gas and benzene phases, while the NO_2_ substituted derivative is the strongest one in water phase.

## 3. Materials and Methods

### 3.1. Computational Details

All calculations were carried out using Gaussian 09 program package [40]. The gas-phase geometries of neutral compounds and the corresponding radical or ionic structures were optimized using the M062X functional without any constraints. The 6-311++G ** basis set including diffuse and polarization functions was used. It allows balanced treatment of neutral compounds and corresponding radicals or ions. The harmonic vibrational frequencies are calculated to confirm that the optimized geometry correctly corresponds to a local minimum, which has only real frequencies. As the solvent would alter the thermodynamically favored mechanism, benzene and water phases were also considered in this work. The solvent effects were conducted by employing the SMD continuum solvent model based on the optimized gas-phase geometries. Single-point electronic energies were then performed at the M062X/6-311+G ** level of theory in different solvent environments based on the optimized geometries.

The reaction enthalpies related to HAT, SET-PT and SPLET mechanisms are usually denoted as follows:

BDE: bond dissociation enthalpy related to HAT. The calculated equation for BDE is:BDE = *H*(ArO•) + *H*(H•) – *H*(ArOH)(6)

IP: ionization potential related to the first step of SET-PT mechanism. The calculated equation for IP is:IP = *H*(ArOH^+^•) + *H*(e^−^) – *H*(ArOH)(7)

PDE: proton dissociation enthalpy related to the second step of SET-PT mechanism. The calculated equation for PDE is:PDE = *H*(ArO•) + *H*(H^+^) – *H*(ArOH^+^•)(8)

PA: proton affinity of phenoxide anion related to the first step of SPLET. The calculated equation for PA is:PA = *H*(ArO^−^) + *H*(H^+^) – *H*(ArOH)(9)

ETE: electron transfer enthalpy related to the second step of SPLET. The calculated equation for ETE is:ETE = *H*(ArO•) + *H*(e^−^) – *H*(ArO^−^)(10)

The calculated gas and solvent phases enthalpies for H^+^, e^−^ and H• were obtained from the references [41,42,43]. The calculated enthalpies for H^+^ in the gas, water and benzene phases are 6.2 kJ/mol, −1050.0 kJ/mol and −870.6 kJ/mol respectively. The calculated enthalpies for e^−^ in the gas, water and benzene phases are 3.1 kJ/mol, −15.1 kJ/mol and −6.6 kJ/mol respectively. The calculated enthalpies for H• in the gas, water and benzene phases are −1306.6 kJ/mol, −1310.6 kJ/mol and −1301.6 kJ/mol respectively.

### 3.2. Statistics Analysis

Pearson correlation is an effective method to study a broad class of relationships among variables [44]. It is based on the covariance matrix of the data and Pearson correlation coefficient is the parameter to evaluate the strength of relationship between two vectors. Normally, Pearson correlation coefficient (Px,y) between two vectors x and y is calculated using the following equation:(11)$$P_{x,y}\;=\;\frac{\sum xy\;-\;\frac{\sum x\sum y}{N}}{\sqrt{(\sum x^{2}\;-\;\frac{\left(\sum x^{2}\right)}{N})\left(\sum y^{2}\;-\;\frac{\left(\sum y^{2}\right)}{N}\right)}}$$
where *N* refers to the size of the signature array and is 11 in this work. The Pearson correlation coefficient is symmetric: *P*_x,y_ = *P*_y,x_. The *P*_x,y_ is among 1 to −1. The more close to 1/−1 the *P*_x,y_, the more related the x and y. The *P*_x,y_ equal to 1 or −1 corresponds to data points lying exactly on a line, or to a bivariate distribution entirely supported on a line.

## 4. Conclusions

Radical scavenging activity of the flavonoid is largely influenced by the structure of the compound. Understanding the role of the substituents is of great importance for the preparation of novel phenolic compounds with an enhanced antioxidant property. In this study, systematic calculations about the influence of the substituents at C3 position on the antioxidant reactivity of naringenin were carried out using DFT calculations.

The most stable geometries of the neutral compounds and the corresponding radical or ionic structures were optimized using the M062X/6-311++G ** method. Based on the optimized geometries, reaction enthalpies (BDE, IP and PA) related to HAT, SET-PT, and SPLET mechanisms have been discussed in detail. Excellent correlations were found between the BDE/IP/PA and Hammett sigma constants. Equations obtained from the linear regression can be useful in the selection of suitable candidates for the synthesis of novel flavonoids derivatives with enhanced antioxidant properties. Besides, it is also found that:

(1) Electron-withdrawing groups enhance BDE and IP, while the electron-donating groups reduce the BDE and IP. The effect of the substituents on PA is on the contrary.

(2) The electronic effect of the substituents on the BDEs of naringenin is mainly governed by the resonance effect, while that on the IPs and PAs of naringenin is mainly controlled by field/inductive effect.

(3) In the gas and benzene phases, HAT is the most favorable mechanism and the antioxidant capacities of naringenin and its derivatives are preferred to present by 4′−OH. In water phase, SPLET is more favored than HAT and SET-PT and the antioxidant capacities are preferred to present by 7−OH.

(4) In the gas and benzene phases, the electron-donating substituents placed at the C3 position would enhance the antioxidant activity of naringenin via weakening the BDE of 4′−OH. In the water phase, they will reduce the antioxidant activity by strengthening the PA of 7−OH. Contrary results occur for the electron-withdrawing group. Besides, the NH_2_ substituted derivative is the strongest antioxidant in the gas and benzene phases, while the NO_2_ substituted derivative is the strongest one in the water phase.

## Figures and Tables

**Figure 1 ijms-20-01450-f001:**
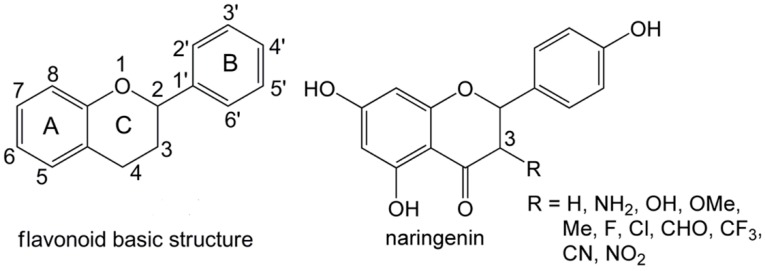
The flavonoid basic structure and the chemical structure of naringenin.

**Figure 2 ijms-20-01450-f002:**
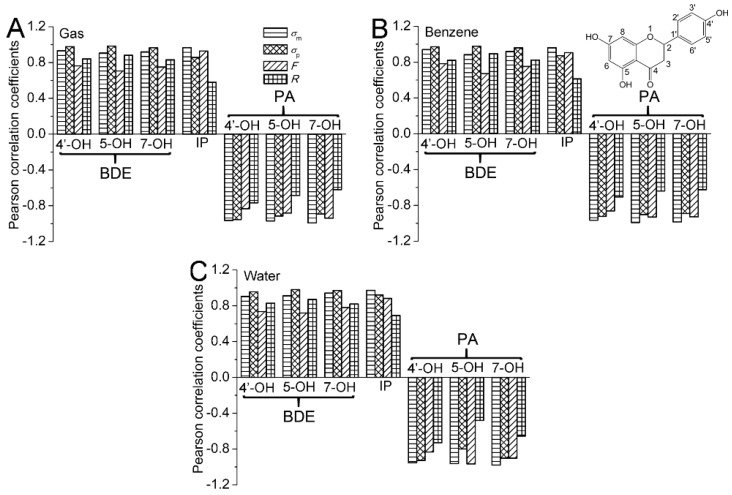
The Pearson correlation coefficients between the Hammett sigma constants (*σ*_m_ and *σ*_p_)/*F*/*R* and the BDE/ ionization potential (IP)/ ionization potential (PA) in the gas (**A**), benzene (**B**) and water (**C**) phases.

**Figure 3 ijms-20-01450-f003:**
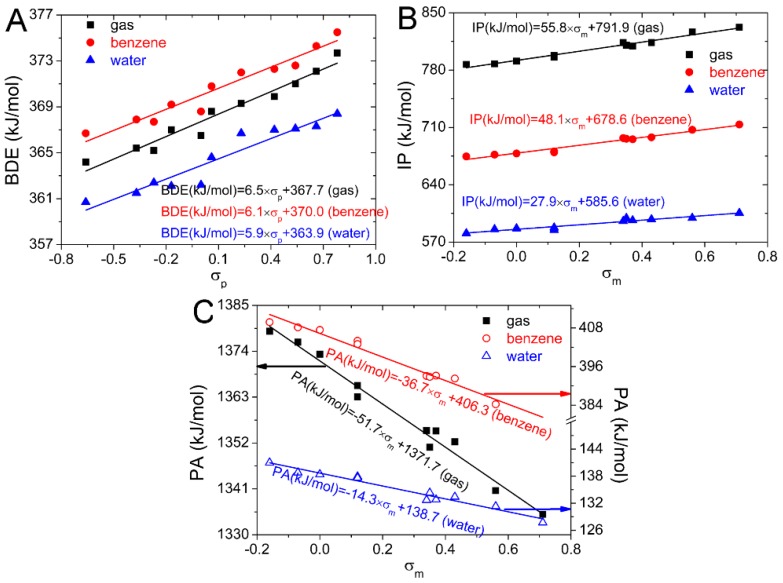
Dependence of BDE (**A**), IP (**B**) and PA (**C)** on Hammett sigma constants for naringenin in gas, benzene and water phases.

**Table 1 ijms-20-01450-t001:** O–H bond dissociation enthalpy (BDE) in kJ/mol obtained by the M062X/6-311+G ** method. The data in the form of the bold and underline represent the lowest BDEs in the molecule.

Substituents	Gas			Benzene			Water		
	4′−OH	5−OH	7−OH	4′−OH	5−OH	7−OH	4′−OH	5−OH	7−OH
H	**366.5**	428	391.1	**368.6**	423.2	395.2	**362.2**	391.2	393.4
NH_2_	**364.2**	425.3	389.7	**366.7**	421.0	393.6	**360.7**	389.9	391.0
OH	**365.4**	426.1	390.1	**367.9**	421.5	393.8	**361.5**	390.1	391.6
OMe	**365.2**	425.8	390.4	**367.7**	421.3	394.2	**362.4**	390.2	392.1
Me	**367.0**	427.2	390.9	**369.2**	422.7	394.5	**362.1**	390.6	392.5
F	**368.6**	429.1	391.5	**370.8**	423.8	396.1	**364.6**	391.6	394.8
Cl	**369.3**	430.1	393.9	**372.0**	423.9	396.6	**366.7**	392.1	395.5
CHO	**369.9**	430.9	394.3	**372.3**	424.1	398.4	**367.0**	392.7	396.0
CF_3_	**371.0**	431.2	394.8	**372.6**	424.7	399.3	**367.1**	393.6	396.9
CN	**372.1**	431.9	396.2	**374.3**	425.5	400.9	**367.3**	393.9	398.5
NO_2_	**373.7**	432.4	397.3	**375.5**	426.1	402.3	**368.4**	394.3	400.5

The bold and underline represent the lowest BDEs in the molecule.

**Table 2 ijms-20-01450-t002:** Ionization potential (IP) in kJ/mol obtained by the M062X/6-311+G ** method.

Substituents	Gas	Benzene	Water
H	791.0	678.2	586.6
NH_2_	786.9	674.6	580.6
OH	797.8	679.6	587.6
OMe	795.7	680.8	584.7
Me	787.7	676.8	585.6
F	813.7	697.1	595.5
Cl	809.3	695.6	596.4
CHO	810.4	696.2	599.2
CF_3_	813.8	698.0	597.7
CN	826.5	707.1	599.4
NO_2_	832.4	713.6	605.4

**Table 3 ijms-20-01450-t003:** Proton affinity (PA) in kJ/mol obtained by the M062X/6-311+G** method. The data in the form of the bold and underline represent the lowest PAs in the molecule.

Substituents	Gas			Benzene			Water		
	4′−OH	5−OH	7−OH	4′−OH	5−OH	7−OH	4′−OH	5−OH	7−OH
H	1406.3	1445.3	**1373.3**	439.2	468.8	**407.4**	162.3	160.2	**138.4**
NH_2_	1409.5	1447.7	**1378.8**	442.2	472.0	**409.8**	164.2	161.9	**141.0**
OH	1405.7	1442.6	**1363.1**	437.9	464.1	**404.0**	162.0	153.8	**137.9**
OMe	1405.5	1435.4	**1365.8**	437.8	462.3	**403.0**	161.9	157.5	**137.6**
Me	1406.9	1447.3	**1376.2**	440.4	471.2	**408.2**	162.6	161.6	**138.8**
F	1404.0	1423.6	**1355.0**	437.8	450.1	**393.1**	161.5	150.0	**132.7**
Cl	1404.6	1424.0	**1354.9**	437.4	451.2	**393.2**	161.2	152.1	**132.8**
CHO	1402.9	1418.9	**1351.0**	436.4	450.0	**392.8**	161.5	152.7	**134.3**
CF_3_	1401.1	1419.0	**1352.3**	436.2	448.5	**392.3**	160.2	151.8	**133.4**
CN	1399.6	1407.0	**1340.6**	433.6	439.4	**384.3**	159.3	148.9	**131.3**
NO_2_	1397.8	1399.6	**1334.9**	431.8	433.0	**379.6**	158.6	142.2	**127.7**

The bold and underline represent the lowest PAs in the molecule.

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
