# Peer review of "DFT Studies on the Antioxidant Activity of Naringenin and Its Derivatives: Effects of the Substituents at C3"

_ijms, 2019, doi:10.3390/ijms20061450_

Round 1
Reviewer 1 Report
The present manuscript is a revised version of a previously submitted manuscript to IJMS (ijms-436504). In this work the authors have performed a DFT study of the substituent effect on the antioxidant activity of the flavonoid naringenin. Several parameters like bond dissociation enthalpy (BDE), Ionization potential (IP) and proton affinity (PA) were calculated for the substituted naringenins and correlated with Hammet constants. I see that the authors have take into account the reviewer suggestions for the manuscript revision. In my opinion, the article is now suitable for publication in the International Journal of Molecular Science.
Minor Revision
In page 3, lines 107-110, the authors wrote:
"The antioxidant activities of naringenin and its C3-OH substituted derivatives kaempferol (4',3,5,7-tetrahydroxyflavanone) have been reported in previous work [31]. It is found that in aqueous phase, the antioxidant activity of kaempferol is stronger than naringenin [31]."
I believe that reference [31] is actually reference [33]. Please check the reference numbering.
Author Response
Thanks for your reminding. We have corrected the reference numbering in the revised namuscript.

Reviewer 2 Report
Comments on manuscript
DFT studies on the
antioxidant activity of naringenin and its derivatives: effects of
the substituents at C3
submitted to I. J.
Mol. Sci.
The authors study
theoretically antioxidant activity of naringenin and its selected
derivatives. The chosen topic is important from a basic and practical
point of view and the research tools are carefully selected.
In addition, the manuscript is written in good English and reads well.
However, there are
some problems needed to be corrected:
1. Figures are too
small and difficult to read (see fig. 2)
2. Labeling of
Figure 1 and substituents in tables (first column) are not clear. A
reader should know directly which derivative is studied.
3. Methodology.
There are close and open shell molecular systems. What kind of
calculations were performed: restricted or unrestricted? What about
spin contamination?
4. Citations: check
abbreviation “Free Radical. Bio. Med.”
In conclusion,
the manuscript needs a minor revisions before its acceptance.
Author Response
The authors study theoretically antioxidant activity of naringenin and its selected derivatives. The chosen topic is important from a basic and practical point of view and the research tools are carefully selected.
In addition, the manuscript is written in good English and reads well.
However, there are some problems needed to be corrected:
Point 1: Figures are too small and difficult to read (see fig. 2)
Response 1: Thanks for your valuable reminding. We have changed the format of Figure 2 and 3 in the revised manuscript (lines 145 and 155).
Point 2: Labeling of Figure 1 and substituents in tables (first column) are not clear. A reader should know directly which derivative is studied.
Response 2: Thanks for your reminding. We have added the substituent types in Figure 1 in the revised manuscript (line 54).
Point 3: Methodology. There are close and open shell molecular systems. What kind of calculations were performed: restricted or unrestricted? What about spin contamination?
Response 3: Thanks. For the close shell systems, we used restricted calculations. For the open shell systems, unrestricted calculations were applied. In this work, we applied the DFT calculation. Since the starting point of the DFT is not to solve the Schrödinger equation in an approximate way, it is meaningless to try to measure the rationality of the wave function from the perspective of spin contamination.
Point 4: Citations: check abbreviation “Free Radical. Bio. Med.”
Response 4: Thanks for your reminding. We have corrected the corresponding abbreviation in the revised manuscript (line 379).

Reviewer 3 Report
The paper entitled “DFT studies on the antioxidant activity of naringenin and its derivatives: effects of the substituents at C3” shows a theoretical approximation of the reaction mechanisms for the free radical scavenging action. The functional and basis set used are adequated to this type of study.
Some major concerns.
a) The introduction section should be improved adding more references about the radical scavenging activity of flavonoids studied by theoretical methods. A lot of references can be found.
b) The authors focused the study on the antioxidant activity of phenolic compounds only for the hydroxyl groups at 4’, 5 and 7 (according to his numbering). To do this, authors considered the HAT, SET-PT and SPLET mechanism. However, the antioxidant activity of flavonoids can be associated to another mechanisms as RAF, SEPT (sequential electron proton transfer) and they should be considered.
c) In page 2, the addition of equation (4) and (5), the two steps of this mechanism, is not equation (3). Authors should revise this mechanism.
d) I am wondering why are you considering exclusively the C3 substituent. I assume that you are using naringenin as a template, but you should mention in the manuscript the substituent effects on other positions could be considered.
e) In materials and methods section you explain that solvent effects have been calculated on optimized gas phase structures. Naringenin structure does not have a large conformational freedom, however the optimized structures, and at the end the descriptors, can change from gas phase to benzene o water. I think you should optimize the structures in the considered solvent. I am know thinking about the different interactions between 5-OH and 4-O in gas phase and in water. This hydrogen bond interaction is obviously affected by the C3 substituent.
f) Finally, in page 9 authors show how the different descriptors are calculated. A comment should be done about the H(H+) and H(e-) values.
Author Response
The paper entitled “DFT studies on the antioxidant activity of naringenin and its derivatives: effects of the substituents at C3” shows a theoretical approximation of the reaction mechanisms for the free radical scavenging action. The functional and basis set used are adequated to this type of study.
Some major concerns.
Point 1: The introduction section should be improved adding more references about the radical scavenging activity of flavonoids studied by theoretical methods. A lot of references can be found.
Response 1: Thanks for your valuable suggestions. We have added more references about the radical scavenging activity of flavonoids studied by theoretical methods in the revised manuscript (references from 31 to 36).
point 2: The authors focused the study on the antioxidant activity of phenolic compounds only for the hydroxyl groups at 4’, 5 and 7 (according to his numbering). To do this, authors considered the HAT, SET-PT and SPLET mechanism. However, the antioxidant activity of flavonoids can be associated to another mechanisms as RAF, SEPT (sequential electron proton transfer) and they should be considered.
Response 2: Thanks for your valuable suggestions. For most of the work related with the antioxidant activity of flavonoids, they analysed the HAT SET-PT and SPLET mechanisms. These three mechanisms are enough to study the antioxidant activity of naringenin and its derivatives in this work. Therefore, in this work, we also analysed the HAT SET-PT and SPLET mechanisms.
Point 3: In page 2, the addition of equation (4) and (5), the two steps of this mechanism, is not equation (3). Authors should revise this mechanism.
Response 3: Thanks for your valuable suggestion. We have corrected this mechanism in the revised manuscript (line 75).
Point 4: I am wondering why are you considering exclusively the C3 substituent. I assume that you are using naringenin as a template, but you should mention in the manuscript the substituent effects on other positions could be considered.
Response 4: Thanks for your valuable reminding. The derivatives of flavonoids with substituents at C3 positions are widely distributed in natural foods such as honey and bee pollens. We have added this statement in the revised manuscript (lines 83 to 85).
Point 5: In materials and methods section you explain that solvent effects have been calculated on optimized gas phase structures. Naringenin structure does not have a large conformational freedom, however the optimized structures, and at the end the descriptors, can change from gas phase to benzene o water. I think you should optimize the structures in the considered solvent. I am know thinking about the different interactions between 5-OH and 4-O in gas phase and in water. This hydrogen bond interaction is obviously affected by the C3 substituent.
Response 5: Thanks. For systems without locally significant ionicity, the solute-solvent interaction has little effect on the structure and there is no need to optimize each solvent separately. Thus, in this work, we optimized the molecular structure in the gas phase and then calculated the single point energy in different solvent phases.
Point 6: Finally, in page 9 authors show how the different descriptors are calculated. A comment should be done about the H(H+) and H(e-) values.
Response 6: Thanks for your valuable suggestions. We have done a comment about the H(H+) and H(e-) values in the revised manuscript (lines 320 to 323).

Author Response.docxRound 2
Reviewer 3 Report
Responses to reviewer's comments are satisfactory to consider the manuscript ready to be published.
The only aspect that must be modified are equations (3) and (5). In equation (3), substitute R- by R (radical). In equation (5), RH- by RH.
Author Response
Point: The only aspect that must be modified are equations (3) and (5). In equation (3), substitute R- by R (radical). In equation (5), RH- by RH.
Response: Thanks for your valuable suggestions. The SET-PT takes place through two consecutive steps (equation (2) and (3)). In equation (3), the R- is the product of equation (2). In equation (5), we have substituted RH- by RH.
This manuscript is a resubmission of an earlier submission. The following is a list of the peer review reports and author responses from that submission.
Round 1
Reviewer 1 Report
There seems to be no particular big problem in publishing the contents of the paper.
